# Systems Genetic Identification of Mitochondrion-Associated Alzheimer’s Disease Genes and Implications for Disease Risk Prediction

**DOI:** 10.3390/biomedicines10081782

**Published:** 2022-07-24

**Authors:** Xuan Xu, Hui Wang, David A. Bennett, Qing-Ye Zhang, Gang Wang, Hong-Yu Zhang

**Affiliations:** 1Hubei Key Laboratory of Agricultural Bioinformaics, College of Informatics, Huazhong Agricultural University, Wuhan 430070, China; xxpinky@webmail.hzau.edu.cn (X.X.); zqy@mail.hzau.edu.cn (Q.-Y.Z.); 2Department of Pathology and Laboratory Medicine, Perelman School of Medicine, University of Pennsylvania, Philadelphia, PA 19104, USA; hui.wang@pennmedicine.upenn.edu; 3Penn Neurodegeneration Genomics Center, Perelman School of Medicine, University of Pennsylvania, Philadelphia, PA 19104, USA; 4Rush Alzheimer’s Disease Center, Rush University Medical Center, Chicago, IL 60612, USA; david_a_bennett@rush.edu; 5Department of Neurological Sciences, Rush University Medical Center, Chicago, IL 60612, USA; 6Hubei Key Laboratory of Central Nervous System Tumor and Intervention, Wuhan 430070, China; wanggang2020@webmail.hzau.edu.cn

**Keywords:** Alzheimer’s disease, mitochondria, association studies in genetics, epistasis, weighted gene co-expression network analysis

## Abstract

Cumulative evidence has revealed the association between mitochondrial dysfunction and Alzheimer’s disease (AD). Because the number of mitochondrial genes is very limited, the mitochondrial pathogenesis of AD must involve certain nuclear genes. In this study, we employed systems genetic methods to identify mitochondrion-associated nuclear genes that may participate in the pathogenesis of AD. First, we performed a mitochondrial genome-wide association study (MiWAS, *n* = 809) to identify mitochondrial single-nucleotide polymorphisms (MT-SNPs) associated with AD. Then, epistasis analysis was performed to examine interacting SNPs between the mitochondrial and nuclear genomes. Weighted co-expression network analysis (WGCNA) was applied to transcriptomic data from the same sample (*n* = 743) to identify AD-related gene modules, which were further enriched by mitochondrion-associated genes. Using hub genes derived from these modules, random forest models were constructed to predict AD risk in four independent datasets (*n* = 743, *n* = 542, *n* = 161, and *n* = 540). In total, 9 potentially significant MT-SNPs and 14,340 nominally significant MT-nuclear interactive SNPs were identified for AD, which were validated by functional analysis. A total of 6 mitochondrion-related modules involved in AD pathogenesis were found by WGCNA, from which 91 hub genes were screened and used to build AD risk prediction models. For the four independent datasets, these models perform better than those derived from AD genes identified by genome-wide association studies (GWASs) or differential expression analysis (DeLong’s test, *p*  <  0.05). Overall, through systems genetics analyses, mitochondrion-associated SNPs/genes with potential roles in AD pathogenesis were identified and preliminarily validated, illustrating the power of mitochondrial genetics in AD pathogenesis elucidation and risk prediction.

## 1. Introduction

Alzheimer’s disease (AD) is the most common neurodegenerative disease. It is estimated that more than 44 million people worldwide have dementia [1], and genome-wide association studies (GWASs) have identified many loci associated with AD. Recently, multiple large-scale GWASs have shown approximately 40 risk loci related to Aβ, tau, immunity and lipid processing [2,3]. Although understanding of the pathogenesis of AD has been progressing for decades, the mechanisms that lead to damage and cognitive defects remain elusive. Thus far, theories of AD etiology include amyloid, tau, and mitochondrial hypotheses [4]. Mitochondrial abnormalities are associated with loss of energy production and synapses, axon transport defects, and cognitive decline [5,6]. Moreover, mitochondrial involvement in AD is supported by neuroimaging features, such as decreased brain glucose and oxygen metabolism and various biological findings, such as altered mitochondrial morphology, impaired respiratory chain function, and mutations in mitochondrial DNA (mtDNA), support this association. In addition, neuronal mitochondrial dysfunction and AD pathology exacerbate each other in a recursive cycle of self-propagation [7]. Therefore, it is of great interest and significance to identify mitochondrion-associated AD disease genes to improve risk prediction in AD pathogenesis.

Mitochondria contain their own small genome of maternal origin that comprises 37 genes encoding: 13 polypeptides, 22 tRNAs, and the small/large rRNA subunits. Due to the limited number of mitochondrial genes, mitochondrial pathogenesis is likely associated with select nuclear genes. Indeed, mitochondrial diseases can be caused by mutations of mtDNA or nuclear genes that encode mitochondrial proteins [8]. Furthermore, signal communication between mitochondria and the nucleus is important for maintaining the homeostasis of the cellular environment [9]. Although interactive mechanisms underlying mitochondrial and nuclear genes largely remain unknown, they may be responsible in part for the slow development of mitochondrion-targeted therapy [10]. To address these issues, we conducted mitochondrial-nuclear genome-wide epistasis analyses, which is a powerful method to infer gene interactions and to provide deeper insights into mitochondrial pathogenesis. To generate a broader view of mitochondrion-based AD pathogenesis in terms of biomolecular networks, a weighted gene co-expression network analysis (WGCNA) was also performed to explore the relationships between different gene sets (modules) and AD clinical features. WGCNA has been successfully applied for various diseases to identify key modules or centrally connected hub genes as potential biomarkers or therapeutic targets [11,12]. Finally, hub genes were screened from WGCNA-derived modules and used to build AD risk prediction models with four independent datasets. Overall, these models exhibited significantly better performance and robustness in predicting AD risk, than those derived from AD genes identified by GWAS or differential expression analysis.

The entire workflow of our analyses is illustrated in Figure 1.

## 2. Materials and Methods

### 2.1. Study Subjects

Genomic and transcriptomic data from the Alzheimer’s Disease Neuroimaging Initiative (ADNI) [13] were used to identify modules and hub genes associated with mitochondrial function in AD. To evaluate the predictive performance of the hub genes, three additional independent datasets were used: the Religious Orders Study and Rush Memory and Aging Project (ROSMAP), GSE5281, and AlzData. ROSMAP are ongoing longitudinal clinical-pathologic cohort studies of ageing and dementia [14]. GSE5281 contains 161 brain samples with gene expression profiles based on Affymetrix Human Genome U133 Plus 2.0 Array (~55,000 transcripts). These samples cover 6 brain regions with approximately 14 biological replicates per brain region, as follows: (1) the entorhinal cortex, (2) the hippocampus, (3) the medial temporal gyrus, (4) the posterior cingulate, (5) the superior frontal gyrus, and (6) the primary visual cortex [15]. AlzData collects a large amount of high-throughput data for public access [16]. It provides cross-platform normalized brain gene expression profiling, including 269 AD and 271 controls from 4 brain regions (EC, the entorhinal cortex; HP, the hippocampus; TC, the temporal cortex; FC, the frontal cortex).

The use of ADNI samples followed the data use agreement at ADNI (https://adni.loni.usc.edu/data-samples/access-data/#access_data, accessed on 23 July 2022). Data from ROSMAP were obtained under a data use agreement with Rush University Medical Center (RUMC). ROS and MAP were approved by the Institutional Review Board of RUMC. All participants provided written informed consent, signed an Anatomic Gift Act, and signed a repository consent allowing their data to be shared. The characteristics of the ADNI, ROSMAP, GSE5281, and AlzData study participants are summarized in Table 1.

### 2.2. Genotyping and Quality Control

The whole mitochondrial genome variant dataset was called from whole genome sequence (WGS) data, including 809 complete and annotated mitochondrial genomes from ADNI. Burrows-Wheeler Aligner (BWA) [17] and Genome Analysis Toolkit (GATK) [18] were used for genome assembly and mutation detection [19]. MITOMAP [20] and Phy-Mer [21] were used for genome annotation [22]. Vcftools [23] was employed to convert ADNI mitochondrial genome variant VCF files into PLINK format. The dataset contains 1604 MT-SNPs. PLINK 1.9 [24] was applied for quality control (QC) and to perform logistic regression to identify susceptibility MT-SNPs associated with AD risk. MT-SNPs were excluded based on the following QC procedures: (1) call rate > 95%; and (2) minor allele frequency (MAF) < 1%. As mitochondrial variation is inherited maternally, the mitochondrial genome does not follow Hardy–Weinberg equilibrium and is not subject to the same threshold as nuclear SNPs [25]. Ultimately, 180 MT-SNPs passed the filtering and QC processes, and candidate MT-SNPs were obtained using a *p* threshold of 0.05.

Autosomal genotype data were obtained from among ADNI Omni2.5M microarray SNP data, which has a total of 812 samples with 2,379,855 SNPs, including the 809 samples for mitochondrial genome data. We performed QC using PLINK 1.9 following common procedures [26]. We filtered out SNPs and individuals based on the following criteria: (1) individual and SNP missingness > 0.02; (2) inconsistencies in assigned and genetic sex of subjects (sex discrepancy); (3) MAF < 0.05; (4) deviations from Hardy–Weinberg equilibrium (HWE) (*p* < 1 × 10^−6^); (5) high or low heterozygosity rates (who deviate ±3 SD from the sample heterozygosity mean rate); (6) relatedness above the threshold (pi-hat > 0.2); and (7) ethnic outliers. Linkage disequilibrium (LD) between SNPs was filtered using the default threshold (*r*^2^ = 0.2).

### 2.3. Mitochondrial Genome-Wide Association Study

We performed a mitochondrial genome-wide association study (MiWAS) to identify MT-SNPs that affect AD risk in ADNI. A logistic regression model was constructed to detect the association between SNPs and AD status (AD vs. no dementia (mild cognitive impairment (MCI)+ cognitive normal (CN))) while controlling for age, sex, and the first 5 principal components (PCs) calculated from the nuclear SNPs with a *p* value less than 0.05. EIGENSTRAT [27] was used for principal component analysis.

### 2.4. Epistasis Screening

For the ADNI mitochondrial genome, PLINK and INTERSNP [28] were used to assess AD-associated epistasis between the mitochondrial and the nuclear genomes. PLINK was first applied to merge the 790 samples that passed QC and shared by the mitochondrial and autosomal datasets. INTERSNP epistasis calculations were performed using the genotypic model containing both additive and dominant effects for 790 AD/no dementia (*n* = 173 vs. 617) and 543 AD/MCI (*n* = 173 vs. 370) individuals. Sex, age (determined by the date of examination in ADNI), and the top five PCs derived among the autosomal SNPs were included in the model as covariates. We deleted MT-nuclear SNP pairs with less than three observed in the cells of the MT-SNP × SNP contingency table, as they may lead to false results. Only MT-nuclear SNP pairs with a cell size greater than 3 or equal to 0 in each cell of the 3 × 3 genotype matrix were retained for further analysis [29]. After filtering, 705,030 valid tests were performed. Therefore, we applied the Bonferroni procedure to assign a rather conservative *p* threshold of 7.09 × 10^−8^ for INTERSNP calculations.

### 2.5. Microarray Data Processing and Differentially Expressed Gene Analysis

The gene expression profile of the blood samples was provided by Bristol-Myers Squibb (BMS), including 743 participants after QC. A NanoDrop and PerkinElmer LabChip GX were used to evaluate the quantity and the quality of the extracted RNA. Affymetrix Human Genome U219 Array (Affymetrix, Inc., Santa Clara, CA, USA), which contains 530,467 probes for 49,293 transcripts was used for expression profiling. The probe set was annotated with the R package “hgu219.db”. All probes were mapped and annotated according to the human genome (GRCh37/hg19).

Uniquely mapped probes with no mismatches were removed. The robust multiarray average (RMA) algorithm was applied to normalize the expression matrix. Then, low-expressed genes were removed, and probes with expression greater than 3 in 10% (~74) of the total samples were retained (density-based filter). After correcting for background (sex, age, education level, batch, RIN), the “limma” R package was used to conduct DEG analysis between different diagnostic statuses (AD vs. no dementia, AD vs. MCI, MCI vs. CN, AD vs. CN). Statistical analysis of microarray data was performed using the function “lmFit” to fit the linear models by weighted or generalized least squares, and the function “eBayes” to generate statistical significance values.

### 2.6. Weighted Co-Expression Network Analysis

The WGCNA R package was used to perform co-expression network analysis with the normalized ADNI matrix [30]. Two outlier samples were removed before analysis. Then, the adjacency matrix was constructed with each entry corresponding to the Pearson’s correlation coefficient (PCC) between each pair of genes. Next, an appropriate soft threshold of eight was selected to construct a scale-free co-expression network from the adjacency matrix. Subsequently, modules were obtained via adaptive branch pruning of hierarchical clustering dendrograms. Branches were merged with a threshold of dissimilarity coefficient <0.2, and 30 modules were obtained.

### 2.7. Identification of Key Modules and Hub Genes

As statistical interactions do not necessarily indicate a biological interaction [31], mitochondrial genes were included to uncover additional genes that biologically interact with mitochondria. Therefore, modules involved in mitochondrial function in AD were identified using the following criteria: first, genes located in mitochondria or with an epistatic effect with mitochondrial genes were enriched in the module; and second, AD-related traits correlated highly with the module. Gene subcellular location information was obtained from the COMPARTMENTS database [32], a web resource updated weekly that integrates evidence on protein subcellular localization. This evidence is derived from a manual literature review, high-throughput screening, automatic text mining, and sequence-based prediction methods. A gene with a score greater than three from the knowledge channel was selected as a reliable mitochondrion localized gene [32]. Epistatic genes were those at the intersection of two INTERSNP calculations, as mentioned above.

After identifying mitochondrial gene-enriched modules, module eigengenes (MEs) were characterized by the first PC of the module expression level. Using the correlation between Mes and sample features to estimate module-feature relationships effectively identifies relevant modules. To further evaluate the correlation strength, we calculated the module significance (MS), which is defined as the average absolute gene significance (GS) of all genes in the module. GS is measured as the log10 conversion of the *p* value (*logP*) in linear regression between gene expression and phenotypic information. Modules with the highest MS score among all modules are those most associated with the phenotype and are selected for further analysis.

Hub genes are highly connected nodes that involve many interactions, and the hub gene in a module may be more important than other genes in the entire network. To identify hub genes in the selected modules, the following three statistics are defined for each gene in WGCNA: (1) connectivity, the degree of connectivity of a gene in the co-expression network is defined as the number of edges connected to the gene; (2) module membership (MM), the MM value can be obtained by analyzing the correlation between the expression of the gene and the first PC of the module (module eigengene); (3) based on GS and MM, the association of each gene with the specified trait and identified genes that have high gene significance and are members of important modules at the same time are selected using the “networkScreening” function. Based on the above three indicators, the following criteria were used to screen for hub genes: GS greater than the mean value of the total GSs within the module, MM > 0.8, and networkScreening.q.weighted < 0.01.

### 2.8. Functional Enrichment Analysis of Key Modules

Key modules were extracted from the network, and enrichment analysis was performed to further explore the functions of each module. The “clusterProfiler” [33] R package was used to perform Gene Ontology (GO) and Kyoto Encyclopedia of Genes and Genomes (KEGG) pathway enrichment analyses. Enrichment results of GO biological process (BP), molecular function (MF), cell component (CC), and KEGG pathway were obtained using *p* < 0.05 and q < 0.2 as the significance threshold.

### 2.9. Construction and Evaluation of the Predictive Model

Among the hub genes in key modules, we selected those that were also DEGs in the ADNI gene expression profile to construct the prediction model. Age and sex were added as covariates. All statistical analyses were performed in R (4.0.2). Three additional independent datasets (ROSMAP, GSE5281, and AlzData) were used for the construction and the evaluation of the model. Considering the number of hub genes, the model was constructed by the random forest algorithm of the “caret” R package, and accuracy was assessed with 10-fold cross validation (10 CV). Afterwards, the “pROC” R package was employed to evaluate and to verify the predictive performance. Associated statistics (including accuracy, sensitivity, and specificity, etc.) were calculated to analyze the prediction results. The importance of variables was based on the model with the highest area under curve (AUC). Furthermore, we conducted survival analysis on the model predictive ability of each dataset by using the “survival” and “survminer” R packages. Samples with predicted values greater than the average level were classified into high-risk groups; otherwise, they were classified into low-risk groups.

## 3. Results

### 3.1. Mitochondrial Genome-Wide Association Study

First, we conducted MiWAS to examine the effects of 1604 MT-SNPs on AD risk in 809 ADNI individuals. Under the logistic regression model, nine MT-SNPs reached statistical significance with a nominal *p*-value < 0.05 (Appendix A). Six of them were located in the mitochondrial control region (m.114C > T, m.152T > C, m.295C > T, m.310T > C, m.462C > T, m.482T > C), which is the main regulatory element for mitochondrial DNA replication and transcription (Figure 2A). A recent study showed that the regulatory factor of the binding sites in the mtDNA control region may be altered in presbycusis, affecting mtDNA gene expression and copy number. Moreover, these variants have potential as diagnostic markers for individuals at a high risk of developing presbycusis [34].

Furthermore, significant loci were annotated based on the mutation information in MITOMAP [35] (Appendix A). Five MT-SNPs (m.114C > T, m.295C > T, m.310C > T, m.462C > T, m.3394T > C) are related to a variety of diseases, such as schizophrenia, bipolar disorder, low VO2max response, glaucoma, and Leber hereditary optic neuropathy (LHON).In particular, *MT-ND1* (m.3394T > C, OR = 2.026, *p* = 0.034), *MT-CO1*(m.6371T > C, OR = 2.380, *p* = 0.015) and *MT- ND5* (m.13966A > G, OR = 2.211, *p* = 0.019) are strikingly elevated in circulating extracellular vesicles (Evs) of MCI and AD patients relative to CN Evs. These mitochondrial genes are involved in encoding critical components of oxidative phosphorylation (OXPHOS). OXPHOS dysfunction can produce ROS and oxidative stress, leading to ageing and neuronal cell death in the AD brain [36]. It has been proposed that these mt-RNAs in plasma Evs can serve as diagnostic and prognostic biomarkers for MCI and AD [37].

### 3.2. Mitochondrial Epistasis Screening

After QC and LD filtering, 790 individuals with 151,497 SNPs were retained for ADNI. Under a conservative Bonferroni-adjusted *p* value (*p* < 7.09 × 10^−8^), m.152T > C (MT-CR) and rs244433 (chr5: 166370007A > G, closest gene: *TENM2*) presented the only significant genetic interaction (*p* = 6.35 × 10^−8^) identified in the AD/no dementia contrast group. *TENM2* (Teneurin Transmembrane Protein 2) is involved in neural development, regulating the establishment of proper connectivity within the nervous system (according to the GeneCards database). The link between *TENM2* and AD is supported by a recent genome-wide diverse meta-analysis [3]. In our study, *TENM2* was found to have a tendency of differential expression in the AD vs. no dementia control group in the GSE5281 dataset (logFC = −0.747, adj.*p* = 0.00366). This result was confirmed using single-cell data from various AD and no dementia samples in the scREAD dataset (https://bmbls.bmi.osumc.edu/scread/, accessed on 23 July 2022). The most significant difference in *TENM2* gene expression was found in excitatory neurons. By querying the GWAS Catalog database (https://www.ebi.ac.uk/gwas/home, accessed on 23 July 2022), the most significant variations linked to *TENM2* are rs6863407, rs35263578, rs2336895, rs4044321, and rs13186288. These variations/risk alleles are related to diseases/phenotypes including gamma-glutamyl transferase, externalizing behavior, smoking initiation, and major depressive disorder. Detailed information can be found in Appendix A.

In addition, 54,942 (AD/no dementia) and 38,643 (AD/MCI) MT-nuclear SNP interactions met the criteria of a cell size either more than three or equal to zero and a nominal *p* threshold of 0.05. Among them, there were 14,340 MT-nuclear SNP interactions replicated in the two contrast groups (Figure 2B, Appendix A). A total of 11,098 SNPs with exact autosomal location information were selected for further analysis by mapping to GRCh37/hg19. These SNPs correspond to 1650 genes, 102 of which are subcellularly located in mitochondria. The top three enriched GO BP terms for the corresponding genes of candidate interactions were axonogenesis (GO:0007409, *p* = 2.94 × 10^−11^), regulation of small GTPase mediated signal transduction (GO:0051056, *p* = 5.28 × 10^−8^), and regulation of cell morphogenesis involved in differentiation (GO:0010769, *p* = 4.80 × 10^−8^). Involvement of synaptic plasticity and axonogenesis markers is highly specific to both tau and AD traits. Overexpression of 4RON human tau in neuroblastoma cells reportedly recruits neurodegeneration-related mitochondria and axonogenesis-related proteins into exosome secretion pathways through different mechanisms [38]. Overall, the results suggest that these SNPs may interact with MT-SNPs associated with AD.

### 3.3. Identification of Differentially Expressed Genes

To remove low-expression genes, 14,057 probes were filtered, and 35,329 probes were obtained (mapped to 15,508 corresponding genes). After correcting for background and under the threshold of *p* < 0.05, there were 2226 DEGs (3068 probes, AD vs. no dementia) (Appendix A), 1505 DEGs (1921 probes, AD vs. MCI) (Appendix A), 986 DEGs (1148 probes, MCI vs. CN) (Appendix A), and 2659 DEGs (3822 probes, AD vs. CN) (Appendix A). To comprehensively screen hub genes, these DEGs were integrated for subsequent analyses. The top 10 DEGs €n the four groups are shown in Appendix A. In addition, there were 34 DEGs under the threshold of adj.*p* < 0.05. These genes with significantly different expression levels were used for subsequent model predictive ability comparison.

### 3.4. Weighted Gene Co-Expression Network Analysis and Detection of Key Modules

To identify gene clusters with different co-expression patterns, which may be related to AD pathologies, gene co-expression network analysis was performed with the WGCNA R package. As a result, 30 modules were obtained after module merging (Figure 3A–C). Subsequently, correlations between different modules and AD-related traits were evaluated by calculating the module significance for each module-trait correlation (Mini-mental State Examination (MMSE), ABETA, TAU, PTAU, AGE, PTETHCAT (ethnicity), PTGENDER (sex), PTRACCAT (race), DIAGNOSIS (CN, MCI, AD), DX (no dementia, AD), APOE2, APOE4). The strongest correlations between modules and traits were as follows: lightsteelblue1—MMSE/AGE (R = 0.103/−0.285 and *p* = 0.005/<0.001); pink—ABETA (R = 0.082 and *p* = 0.003); thistle2—TAU/PTAU (R = 0.094/0.099 and *p* = 0.011/0.007); violet—AGE (R = 0.272 and *p* < 0.001); plum1—PTGENDER (R = 0.99 and *p* = 0); darkgreen—DX (R = 0.127 and *p* < 0.001); and cyan—APOE4 (R = 0.075 and *p* = 0.041) (Figure 3D).

Furthermore, 1,474 mitochondrion-located genes were enriched in 11 modules (hypergeometric test, *p* < 0.05), and 15 modules were enriched by 11,098 SNPs obtained by mitochondrial epistasis screenings (hypergeometric test, *p* < 0.05, false discovery rate (FDR) q < 0.05) (Appendix A). Among these mitochondrion-relevant gene-enriched modules, six were selected for subsequent analyses and showed the strongest associations with AD-related traits (lightsteelblue1-MMSE/APOE2, pink-ABETA, thistle2-PTAU, violet-AGE, darkgreen-DX, and cyan-APOE4) (Figure 3D and Appendix A).

### 3.5. Functional Enrichment Analysis of Key Modules

GO and KEGG analyses were then performed to explore the functions for the genes clustered in the six modules. The KEGG pathway enrichment results demonstrated the genes in the lightsteelblue1 module to be primarily enriched in pathways associated with the NF-kB signaling pathway, necroptosis, and apoptosis, among others. The results of GO BP analysis indicated that genes in the lightsteelblue1 module primarily regulate T cell differentiation and mitochondrial depolarization. According to GO BP and KEGG terms, the darkgreen module showed significant enrichment in many immune responses, such as B cell activation, the antigen receptor-mediated signaling pathway, and the B cell receptor signaling pathway. Genes in the thistle2 module were enriched in several biological processes of the nervous system, including calcium ion homeostasis and axon and neuron projection regeneration. Furthermore, KEGG and GO BP MF enrichment results demonstrated that the pink module was significantly associated with ubiquitin mediated proteolysis and oxygen transport or metabolic processes, which are all closely associated with accumulation of amyloid Aβ, oxidative stress, and other pathological processes of AD. Detailed information about the top 10 terms of each category for the six modules is presented in Appendix A, and Appendix A.

### 3.6. Identification of Hub Genes

Based on the criteria described above, hub genes in six mitochondrial/AD trait-related modules were screened out (lightsteelblue1-MMSE/APOE2: 3, pink-ABETA: 170, thistle2-PTAU: 1, violet-AGE: 3, darkgreen-DX: 31, and cyan-APOE4: 51). Through comparison with DEGs, 91 hub genes and DEGs were selected for further analyses (Appendix A). There were 16 genes at the intersection of these hub genes and epistatic genes from our previous screening (*ACSL1*, *AFF3*, *APP*, *BACH2*, *BCL11A*, *DCAF12*, *DOCK5*, *NEDD4L*, *OSBP2*, *PSMF1*, *PTPRE*, *RASSF2*, *SIRPA*, *SLC6A6*, *ST6GAL1*, *TNS1*). In addition, 10 genes were found at the intersection of hub genes and genes located in the mitochondria from the COMPARTMENTS database (*APP*, *TRAK2*, *BCL2L1*, *LGALS3*, *ABHD5*, *MXD1*, *ACSL1*, *STAP1*, *GLRX5*, *SNCA*). These hub genes had only one intersection with the well-known AD pathogenic genes (*PICALM*) [2] (Appendix A).

### 3.7. Construction and Evaluation of the Predictive Model

To explore the relationship between hub genes and AD risk, the random forest algorithm was applied to the 743 ADNI cohort to establish a predictive model. In addition, the predictive power of the hub genes was compared between ADNI DEGs and the recognized AD pathogenic genes [2] (Appendix A). *APOE* is the most important genetic factor of AD and accounts for approximately 5–9% of heritability [39]; thus, the prediction efficiency of hub genes was compared with recognized pathogenic genes other than *APOE*. ROC curves showed that the AUC of hub genes (Hub Genes AUC: 0.623) was significantly higher than that of AD pathogenic genes and AD pathogenic genes excluding *APOE* (Sig Genes AUC: 0.541, Sig~*APOE* AUC: 0.528; DeLong’s test, *p* = 0.02/0.008) (Figure 4A). There was no significant difference in AUC between hub genes and ADNI DEGs (adj.*p* < 0.05) (DEGs AUC: 0.634; DeLong’s test, *p* = 0.9).

The prediction power of hub genes was verified in three other independent brain sample datasets. Among 542 subjects in ROSMAP, the AUC of hub genes was 0.737 (Hub Genes AUC: 0.737). However, the AUC of AD pathogenic genes was only approximately 0.65 (Sig Genes AUC: 0.646, Sig~*APOE* AUC: 0.659; DeLong’s test, *p* = 0.01/0.03), and DEGs had a minimum AUC of 0.561, which was significantly lower than that of the hub genes (DeLong’s test, *p* = 4 × 10^−4^) (Figure 4B). For 161 AD and normal aged brain samples in GSE5281, the AUC for hub genes was greater than 0.9 (Hub Genes AUC: 0.922). Although the AUC of the hub genes was higher than that of the AD pathogenic genes, it did not show a significant improvement due to the limited samples, (Sig Genes AUC: 0.889, Sig~*APOE* AUC: 0.899; DeLong’s test, *p* = 0.3/0.4). However, the AUC of hub genes was significantly improved compared to that of the DEGs (DEGs AUC: 0.922; DeLong’s test, *p* = 0.002) (Figure 4C).

Regarding integrated data from AlzData in four brain regions, the lowest AUC was 0.610 (for the FC region) because the expression values of 2/3 of the hub genes in this region were missing. For the other three brain regions, HP, EC, and TC, AUC values reached 0.757, 0.788, and 0.808, respectively (Figure 5A).

Furthermore, Kaplan–Meier survival curves illustrated that individuals in the high-risk group had a significantly earlier age of AD onset, as found for all ADNI, ROSMAP, GSE5281, and AlzData datasets (Log-rank test, *p* < 0.0001 (excluding AlzData_TC *p* = 0.0006)) (Figure 4D–F and Figure 5B–E).

### 3.8. Identification of Critical Genes

To determine the overall importance of hub genes, the gene importance of the random forest models in the above-mentioned seven different datasets were integrated (Appendix A). The top 10 hub genes were *RTN3*, *RASSF2*, *TCL1A*, *BCL11A*, *RANBP10*, *REPS2*, *VCAN*, *TMCC3*, *EPB41*, and *NEDD4L*, which may play critical roles in the pathologies of AD through interaction with mitochondrial genes or participation in mitochondrion-related biological processes. Indeed, eight of them are supported by literature evidence related to mitochondria, and seven genes are reported to be related to AD (Table 2).

AlzData was used to further characterize and confirm the expression patterns of these 10 critical genes. Indeed, we found that these genes have significant up- or down-regulated patterns (FDR < 0.05) in different areas of the brain, except for *EPB41*. For example, *RTN3* showed significant down-regulation in EC, HP, and TC, with TC being the most significant (log2 FC = −0.83, FDR = 0.0003). Similarly, *NEDD4L* displayed significant down-regulation in the TC region (log2 FC = −0.48, FDR = 0.0001). In contrast, we observed a significant upregulation of *VCAN*, *TMCC3*, and *SLC14A1* in TCs (log2 FC = 0.7/0.51/1.4, FDR = 0.001/0.002/0.0003, respectively) (Appendix A).

## 4. Discussion

We first performed a mitochondrial genome-wide association analysis of AD. Under a *p* threshold of 0.05, 9 potential MT-SNPs related to AD risk were detected. Previous studies support many of the MT-SNPs identified by our analysis, including m.114C > T, m.295C > T, m.310C > T, m.462C > T, and m.3394T > C. In particular, m.114C > T and m.3394T > C have been confirmed to be related to a variety of mental diseases including AD (Appendix A). Then, we screened epistatic loci between the mitochondrial genome and the nuclear genome. We obtained 14,340 MT-nuclear SNP interactions under a *p* threshold of 0.05 of which 11,098 SNPs with accurate autosomal location information were used for subsequent analyses (Appendix A).

Furthermore, through analysis of ADNI gene expression data, we found DEGs in four contrast groups under the threshold of *p* < 0.05 (Appendix A, Appendix A), reflecting the RNA expression changes in AD blood samples. The top-hit up-regulated genes in the AD vs. CN contrast group, such as *CREB5* (cAMP responsive element binding protein 5), *MAPK14* (mitogen-activated protein kinase 14), and *CD63* (CD63 molecule), are similar to the results of a recent genome-wide transcriptome analysis [40].

WGCNA was used to construct a co-expression network and to detect gene modules in the normalized matrix. As a result, 30 modules were obtained after module merging (Figure 3A–C). Afterward, correlations between the modules and AD-related traits were determined (Figure 3D). Among them, 11 modules were enriched with 1474 mitochondrion-located genes; 15 modules were enriched with 11,098 autosomal SNPs with potential interactions with 9 AD-related MT-SNPs (Appendix A). Based on the above findings, we selected for further investigation the six modules that had the strongest association with AD-related traits and that were simultaneously enriched by mitochondrion-related genes (lightsteelblue1-MMSE/APOE2, pink-ABETA, thistle2-PTAU, violet-AGE, darkgreen-DX, and cyan-APOE4).

GO and KEGG pathway analyses are key to understanding disease mechanisms. In addition to direct enrichment of many neurological-related biological processes in the thistle2 module, many other modules also showed enriched terms related to AD pathologies. For example, the GO BP results of the lightsteelblue1 module indicated that this module is related to immune processes such as T cell differentiation and mitochondrial depolarization, which may be linked to factors such as neuroinflammatory response and the combination of tau and amyloid-β protein precursor (APP) that lead to the damage of mitochondrial phagocytic function [41,42]. KEGG pathway analysis demonstrated that the lightsteelblue1 module is enriched in the NF-κB signaling pathway, necroptosis, and apoptosis, among others. NF-κB activity is increased in most nerve cells, such as neurons, microglia and astrocytes, when acute or chronic neurological diseases occur [43] (Appendix A, Appendix A).

Overall, we screened 91 hub genes from the 6 modules, which were also significant DEGs (Appendix A). *APP* was the only gene at the intersection of hub genes, epistatic genes, and mitochondrial located genes. Studies have shown that accumulation of mutated APP and Aβ in the hippocampus is responsible for abnormal mitochondrial dynamics and defective biogenesis [44]. As defective mitochondrial homeostasis plays a pivotal role in the pathogenesis of AD, targeting mitochondrial dysfunction by offsetting the early accumulation of APP may be a promising therapeutic intervention for AD [45]. Besides, many investigations have verified the association between APP and a mitochondrial enzyme, amyloid binding alcohol dehydrogenase (ABAD). One study used a yeast two-hybrid system in which ABAD interacts with Aβ and is present in the mitochondrial matrix. Immunoprecipitation studies in the brain have shown the formation of complexes between ABAD and Aβ in brain and mitochondrial extracts [46]. Aβ–ABAD interaction in mitochondria reduces enzymatic function, increases the production of reactive oxygen species (ROS), and affects energy consumption. This process has also been proven in AD mouse models [47,48]. Therefore, interception of the Aβ-ABAD interaction has the potential to protect against AD pathological processes [46,49]. Additionally, 37 genes among all hub genes have been reported to be associated with AD, and 10 genes appear in more than 10 related literature reports (*APP*, *SNCA*, *PICALM*, *BCL2L1*, *RTN3*, *GCA*, *SLC2A1*, *LGALS3*, *FPR2*, and *NUMB*). A total of 15 hub genes have research records related to MCI, including 4 genes with more than 10 publications (*APP*, *GCA*, *PICALM*, and *CD22*). The difference between the above two gene sets, CD22, which is known to modulate immune system activation, has also been discovered to be a negative regulator of microglia phagocytosis in the brain. CD22 inhibition increases microglial phagocytosis of Aβ oligomers and ameliorates cognitive impairment in elderly mice [50]. According to a recent study, higher plasma soluble CD22 (sCD22) levels are associated with accelerated cognitive deterioration. Furthermore, plasma sCD22 levels predicted longitudinal cognitive deterioration during 7.5 years of follow-up [51]. This evidence suggests that *CD22* may be an important marker to help us identify patients who are more likely to transform from MCI to AD. In addition, 44 genes have been reported to be associated with MT of which 12 (*BCL2L1*, *APP*, *SNCA*, *SLC2A*, *NAMPT*, *LGALS3*, *ACSL1*, *TRAK2*, *GCA*, *GLRX5*, *FPR2,* and *ABHD5*) are included in more than 10 reports.

Moreover, KEGG pathway enrichment of these hub genes revealed some immune processes/diseases, such as intestinal immune network for IgA production (hsa04672, adj.*p* = 5.77 × 10^−43^), type I diabetes mellitus (hsa04940, adj.*p* = 1.21 × 10^−42^) and inflammatory bowel disease (hsa05321, adj.*p* = 7.41 × 10^−42^), which are related to imbalance of the gut microbiota [52], and emerging evidence shows that AD may begin in the gut, and it is closely related to such an imbalance [53]. Overall, our analyses suggest that mitochondrial dysfunction may induce AD, affecting the gut microbiota.

Interestingly, the prediction results showed that the AUCs for the hub genes in the ADNI and ROSMAP datasets were significantly higher than those for recognized AD pathogenic genes (Appendix A). When compared with ADNI DEGs (adj.*p* < 0.05), the AUCs derived from hub genes in the ROSMAP and GSE5281 datasets were significantly improved, except for ADNI gene expression data (Figure 4), showing that the hub genes identified by systems genetics methods are more robust at predicting AD risk. This point was further validated using the AlzData database, through which we observed accurate AD risk prediction of hub genes in four regions of the brain (Figure 5).

To identify critical genes in the mechanism of AD and mitochondria, we obtained the top 10 hub genes based on the synthetic importance of hub genes in all models (Appendix A, Appendix A). Most of them have been reported to be associated with AD and/or mitochondria (Table 2). Among them, the top-ranked gene *RTN3* (Reticulon 3) has been shown to be involved in the development of neurodegenerative diseases, especially in AD (according to the GeneCards database). *RTN3* belongs to the reticulon family which interacts with β-amyloid converting enzyme 1 (BACE1) and regulates its activity, as well as production of Aβ [54]. In addition, *RTN3* is involved in the process of mitochondrial specific autophagy, which has been proven to be beneficial for inflammatory diseases by eliminating damaged mitochondria and maintaining homeostasis [55]. Moreover, the transcriptional profiles of top 10 hub genes in human brain development between different genders were explored using BrainSpan (http://www.brainspan.org, accessed on 23 July 2022) and GenTree [56] databases (Appendix A). It can be seen that the expression levels of these genes differ significantly between the genders at different developmental stages. For example, the expression levels of *TCL1AM*, *RANBP10,* and *EPB41* peaked in the female subcortical region at 34 post-conception week (PCW) and were significantly higher than those of males at the same period. With aging, the expression levels of *RASSF2*, *RABP10*, *REPS2*, and *TMCC3* were higher in males than in females in the same brain tissue regions, and the disparities tended to grow. These results may help to explore gender differences in the onset of AD.

## 5. Limitations

There may be some possible limitations in this study. Studies to date explaining the association between mitochondrial genetics and AD risk have mostly used incomplete sequence data and/or very small sample sizes [22]. In our study, we used a dataset of 809 annotated whole mitochondrial genomes. Although the sample size was increased compared with previous studies, it may still be insufficient in genome research. This is probably the reason most of our findings only reached a nominally significant threshold. In addition, in order to preserve consistency in quality control and screening criteria across all available samples, we did not include mitochondrial haplotypes or SNPs reported in other studies that are associated with AD, which may have resulted in incomplete findings in certain circumstances. The development of a larger dataset would facilitate research in this important area.

Moreover, in the risk prediction models, we simply considered hub genes from six AD-related modules that were also enriched by mitochondrion-related genes. Therefore, some important pathogenic genes related to AD pathology, but not closely related to mitochondria may have been missed. A more comprehensive assessment of AD pathogenic genes may improve the prediction accuracy and provide a more profound understanding of the pathological mechanism of AD.

## 6. Conclusions

Accumulating evidence shows that mitochondrial dysfunction contributes to the ageing process, thereby increasing the risk of AD. Due to the limited number of mitochondrial genes, certain nuclear genes must play critical roles in the mitochondrial pathogenesis of AD. Although, the interactive mechanisms underlying mitochondrial and nuclear genes remain elusive [57], they may be responsible for the slow development of mitochondrial medicine. Here, through mitochondrial systems genetic analyses, mitochondrion-associated nuclear SNPs/genes with important roles in the pathogenesis of AD are identified and preliminarily validated. To a certain extent, our research underscores the essential role of mitochondrial dysfunction in the pathogenesis of AD and provides new insight into the unclear mitochondrion-relevant pathogenic mechanisms for AD. Moreover, we reveal that mitochondrion-associated nuclear genes have important implications for AD risk prediction and AD mitochondrial medicine.

## Figures and Tables

**Figure 1 biomedicines-10-01782-f001:**
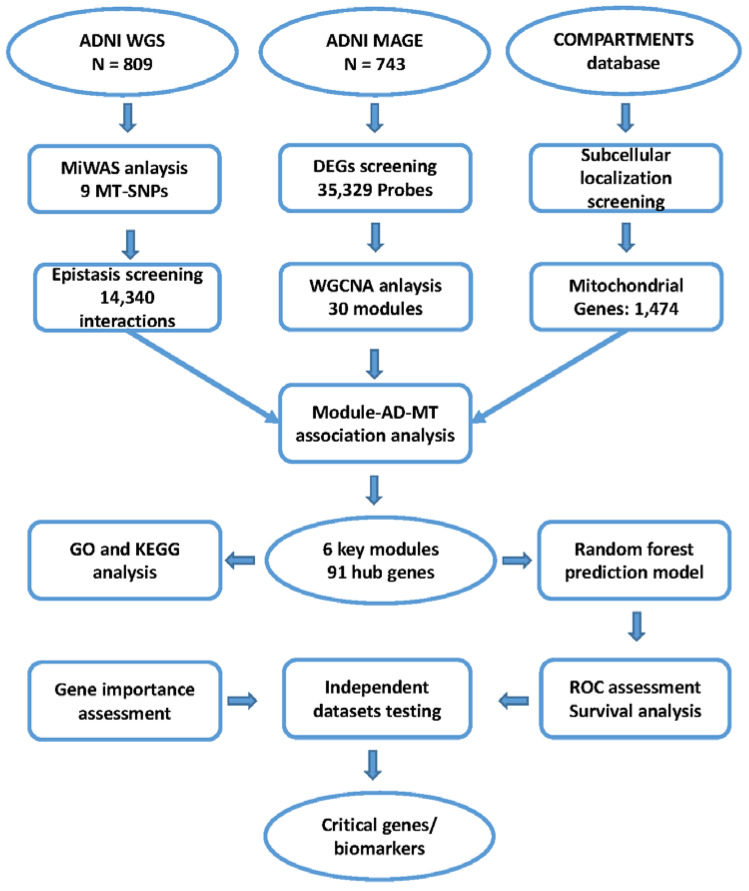
The workflow of analysis and validation procedures. Abbreviations: AD, Alzheimer’s disease; ADNI, Alzheimer’s Disease Neuroimaging Initiative; DEG, differentially expressed gene; SNP, single nucleotide polymorphism; MAGE, microarray and gene expression; MiWAS, mitochondrial genome-wide association study; MT, mitochondria; WGCNA, weighted co-expression network analysis; GO, gene ontology; KEGG, Kyoto Encyclopedia of Genes and Genomes.

**Figure 2 biomedicines-10-01782-f002:**
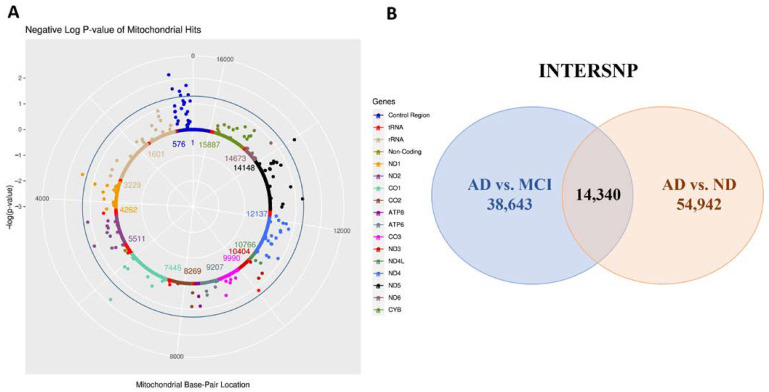
MiWAS result and epistasis screening. (**A**) Mitochondrial solar plot of MiWAS result. Nine MT-SNPs with a nominal *p*-value < 0.05 were considered statistically significant associates with AD risk (Appendix A). Six of them are located in the control region of the mitochondria. (**B**) Epistasis screening of nine MT-SNPs significantly related to AD risk. A total of 54,942 (AD/no dementia) and 38,643 (AD/MCI) MT-nuclear SNP interactions met the criteria of a cell size either more than 3 or equal to 0 and a nominal *p* threshold of 0.05. Among them, there were 14,340 MT-nuclear SNP interactions replicated in the 2 contrast groups (Appendix A).

**Figure 3 biomedicines-10-01782-f003:**
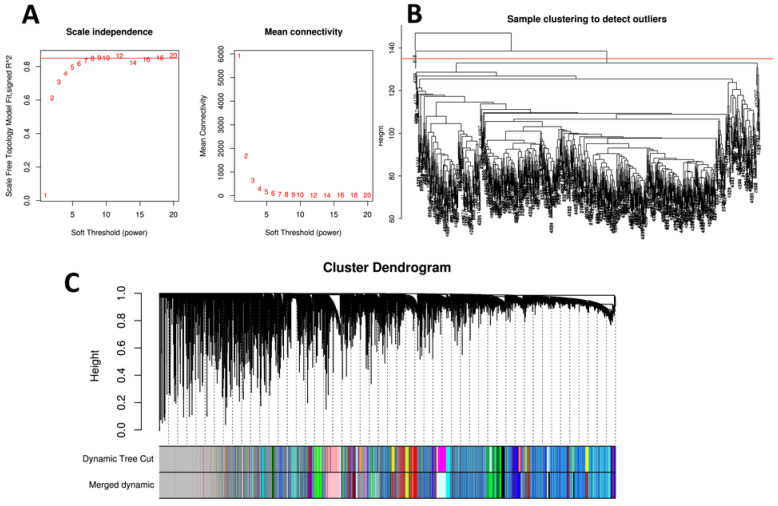
The workflow for WGCNA and identification of key modules. (**A**) Analysis of the scale-free fit index and the mean connectivity for various soft-thresholding powers (β). The approximate scale-free topology can be attained at β = 8. (**B**) Cluster analysis of samples to detect outliers. (**C**) Cluster dendrogram: Each color represents one specific co-expression module, and black branches represent genes. (**D**) Heatmaps of the correlation between eigengene and AD-related traits (MMSE, ABETA, TAU, PTAU, AGE, PTETHCAT (ethnicity), PTGENDER (sex), PTRACCAT (race), DIAGNOSIS (CN, MCI, AD), DX (no dementia, AD), APOE2, APOE4). Each row corresponds to a module eigengene, and each column corresponds to a clinical AD trait. Each cell contains the corresponding correlation and *p*-value.

**Figure 4 biomedicines-10-01782-f004:**
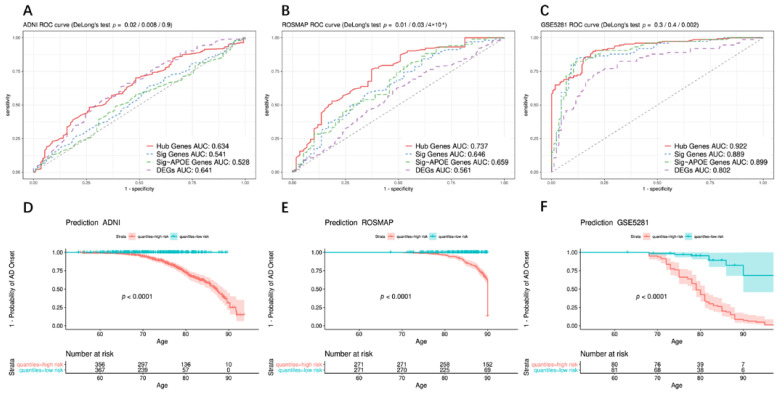
Construction and evaluation of AD predictive models in different independent datasets (ADNI, ROSMAP, GSE5281). Random forest algorithm was applied on the three different cohorts to establish AD predictive models through 10-fold CV (ADNI (**A**), ROSMAP (**B**), GSE5281 (**C**)). DeLong’s tests were used to compare the difference between ROC curves. Kaplan–Meier survival curves were used to estimate the accuracy of the hub genes signature on predicting survival. Individuals were well stratified by their risk predictions, and individuals in the high-risk group had a significantly earlier age of AD onset, as found for all ADNI (**D**), ROSMAP (**E**), GSE5281 (**F**) datasets (Log-rank test, *p* < 0.0001).

**Figure 5 biomedicines-10-01782-f005:**
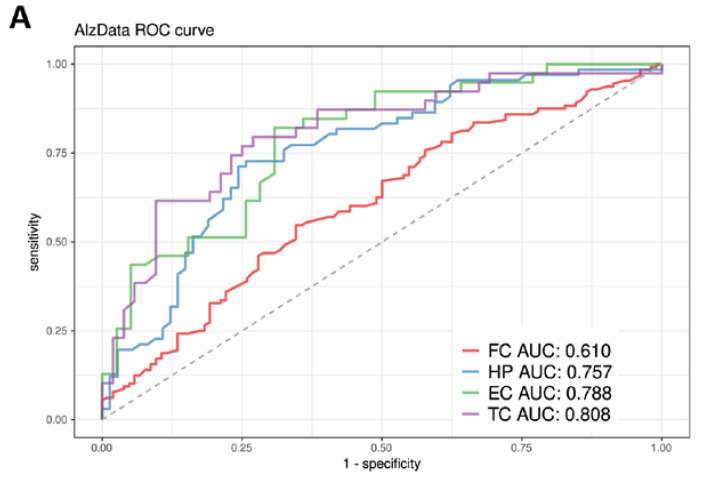
Construction and evaluation of AD predictive models based on AlzData. (**A**) Performance of predictive models based on AlzData. Due to the lack of expression values of approximately 2/3 hub genes, the FC region had the lowest AUC (0.610). For the other three regions, HP, EC, and TC, AUC values reached 0.757, 0.788, and 0.808, respectively. Kaplan–Meier survival curves showed that individuals in the high-risk group exhibited a significantly earlier age of AD onset in EC (**B**), HP (**C**), TC (**D**), and FC (**E**) (Log-rank test, *p* < 0.0001 (excluding AlzData_TC *p* = 0.0006)). Abbreviations: EC, Entorhinal Cortex; HP, Hippocampus; TC, Temporal Cortex; FC, Frontal Cortex.

**Table 1 biomedicines-10-01782-t001:** Characteristics of study participants from ADNI mitochondrial genome cohort (*n* = 809), ROSMAP (*n* = 542), GSE5281 (*n* = 161), and AlzData (*n* = 540).

**ADNI**	**AD (*n* = 175)**	**ND (*n* = 634)**	**Diff (*p*) ***
Sex (F/M)	70 F, 105 M	293 F, 341 M	0.15
Age, y (SD)	73.95(7.66)	73.04 (6.90)	0.13
**ROSMAP**	**AD (*n* = 220)**	**ND (*n* = 322)**	**Diff (*p*) ***
Sex (F/M)	147 F, 73 M	128 F, 194 M	<0.0001
Age, y (SD)	88(3.51)	85.5 (5.00)	<0.0001
**GSE5281**	**AD (*n* = 87)**	**ND (*n* = 74)**	**Diff (*p*) ***
Sex (F/M)	37 F, 50 M	21 F, 53 M	0.07
Age, y (SD)	79.8 (6.91)	79.5 (8.92)	0.8
**AlzData (EC)**	**AD (*n* = 39)**	**ND (*n* = 39)**	**Diff (*p*) ***
Sex (F/M)	18 F, 21 M	17 F, 22 M	1.00
Age, y (SD)	82.4 (7.38)	78 (11.1)	0.04
**AlzData (HP)**	**AD (*n* = 74)**	**ND (*n* = 66)**	**Diff (*p*) ***
Sex (F/M)	45 F, 29 M	23 F, 43 M	0.002
Age, y (SD)	83.1 (9.44)	80.2 (9.68)	0.07
**AlzData (TC)**	**AD (*n* = 52)**	**ND (*n* = 39)**	**Diff (*p*) ***
Sex (F/M)	14 F, 20 M *	18 F, 21 M*	0.81
Age, y (SD)	83.1 (9.44)	80.2 (9.68)	0.07
**AlzData (FC)**	**AD (*n* = 104)**	**ND (*n* = 128)**	**Diff (*p*) ***
Sex (F/M)	44 F, 46 M *	55 F, 65 M *	0.68
Age, y (SD)	84.7 (7.53)	81.7 (10.60)	0.01

Abbreviations: ROSMAP, the Religious Orders Study/the Rush Memory and Aging Project; ADNI, Alzheimer’s Disease Neuroimaging Initiative; AD, Alzheimer’s disease; ND, no dementia; Diff, statistical difference between AD and ND; EC, Entorhinal Cortex; HP, Hippocampus; TC, Temporal Cortex; FC, Frontal Cortex; F, female; M, male; SD, standard deviation. * Missing values exist.* *p* values are calculated by Fisher’s exact tests (for sex) or two-sample *t*-tests (for age at death, age at AD, and age).

**Table 2 biomedicines-10-01782-t002:** Summary information of top 10 critical genes.

Gene	Describe	Compartment *	GO BP *	BioSystems Pathway *	AD Literature (PMID)	MT Literature (PMID)
RTN3	Reticulon 3	plasma membrane; endoplasmic reticulum	GO:0006915; GO:0016032; GO:0071786; etc.	Pathways of neurodegeneration-multiple disease (Alzheimer disease); Transmission across Chemical Synapses (Neuronal System)	23827971; 29356939; 28733667; etc.	32048886; 17191123; 17031492; etc.
RASSF2	Ras Association Domain Family Member 2	nucleus	GO:0001501; GO:0001503; GO:0006468; etc.	Hippo signaling pathway-multiple species	——	22674380; etc.
TCL1A	TCL1 Family AKT Coactivator A	nucleus	GO:0007275; GO:0008284; GO:0010918; etc.	PI3K/Akt Signaling	——	26041471; 10983986; 30282833; etc.
BCL11A	BAF Chromatin Remodeling Complex Subunit BCL11A	nucleus	GO:0000122; GO:0006357; GO:0010976; etc.	——	30180184; 33911114; etc.	33091040; 27838552; etc.
RANBP10	RAN Binding Protein 10	cytosol	GO:0005515; GO:0031267	Signaling events mediated by Hepatocyte Growth Factor Receptor (c-Met)	28659384; 28744327; etc.	——
REPS2	RALBP1 Associated Eps Domain Containing 2	cytosol	GO:0006897; GO:0007173; GO:0016197; etc.	EGF/EGFR Signaling Pathway	32597797; etc.	——
VCAN	Versican	extracellular	GO:0001501; GO:0007417; GO:0007155; etc.	Direct p53 effectors; Regulation of Wnt-mediated beta catenin signaling and target gene transcription; Spinal Cord Injury	7793988; 29752348; 28724990; etc.	30622695; 29060675; etc.
TMCC3	Transmembrane And Coiled-Coil Domain Family 3	Endoplasmic; reticulum	——	——	——	——
EPB41	Erythrocyte Membrane Protein Band 4.1	plasma membrane; nucleus; cytosol	GO:0007049; GO:0008360; GO:0030036; etc.	Syndecan-2-mediated signaling events; Neuronal System	22815752; 24718034; etc.	——
NEDD4L	NEDD4 Like E3 Ubiquitin Protein Ligase	Nucleus; cytosol	GO:0000122; GO:0000209; GO:0003254; etc.	Neurotrophic factor-mediated Trk receptor signaling; Ubiquitin mediated proteolysis TGF-beta Signaling Pathway	32140098; 27686364; 28377502; etc.	31959741; 32140098; etc.

Abbreviations: AD, Alzheimer’s disease; MT, mitochondria. * Basic information comes from https://www.genecards.org, accessed on 23 July 2022.

## Data Availability

The genetic data and gene expression data for co-expression network construction can be applied from the ADNI website (http://adni.loni.usc.edu, accessed on 23 July 2022). The gene expression profiles and post-mortem pathological measurements of AD patients from ROSMAP can be obtained through RADC (https://www.radc.rush.edu, accessed on 23 July 2022) or the ADKnowledge Portal (www.synapse.org, accessed on 23 July 2022). The microarray dataset GSE5281 and the corresponding clinical information can be downloaded from the Gene Expression Omnibus (GEO) database of the NCBI database (https://www.ncbi.nlm.nih.gov/, accessed on 23 July 2022). The AlzData database can be accessed through http://www.alzdata.org/index.html (accessed on 23 July 2022). The COMPARTMENTS database can be accessed through https://compartments.jensenlab.org/ (accessed on 23 July 2022). The scREAD database can be accessed through https://bmbls.bmi.osumc.edu/scread/ (accessed on 23 July 2022). The GenTree database can be accessed through http://gentree.ioz.ac.cn/index.php (accessed on 23 July 2022). The BrainSpan database can be accessed through http://www.brainspan.org/ (accessed on 23 July 2022).

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
