# Peer review of "Systems Genetic Identification of Mitochondrion-Associated Alzheimer’s Disease Genes and Implications for Disease Risk Prediction"

_biomedicines, 2022, doi:10.3390/biomedicines10081782_

Round 1

Reviewer 1 Report

The article of Xu and co-workers is well stritten and very interesting. Mitochondria in AD pathogenesis are important, considering the recent theory about the oxidative stress as precursor event of Ab or Tau aggregation. 

I have only a consideration, in your analysis have you any idea about ABAD genes? amyloid-binding alcohol dehydrogenase (ABAD) is a mitochondrial protein involved in Ab cytotoxicyty. 

Moreover, could be interesting to compare the data also in terms of male and female sex, taking in mind the difference of AD onset, in particular the major female susceptibility induced by estrogen depletion during aging. 

Author Response

We sincerely thank this reviewer for reading our paper carefully and giving comments and suggestions, which have helped us to improve our research.

To address reviewer’s consideration about ABAD genes. We first checked the GeneCards database (https://www.genecards.org/) and found that the gene alias was HSD17B10. According to the COMPARTMENTS database, HSD17B10 is indeed a mitochondrial located gene and is included in the mitochondrial related genes we considered. We then tried to explore the expression pattern of this gene in the datasets we used, such as ADNI, ROSMAP, GSE5281, AlzData, scREAD. Unfortunately, there was no differential expression information associated with ABAD or HSD17B10. Therefore, this gene may show a modest contribution in our subsequent analyses. To further explore the potential contribution of genes associated with ABAD, we used the Harmonizome database (https://maayanlab.cloud/Harmonizome/) [1] to retrieve the most relevant genes for ABAD genes in a collection of 112 datasets of gene and protein information. According to GeneRIF Biological Term Annotations, APP (Amyloid Beta precursor protein) was found to appear in literature-supported statements, along with the biological term ABAD. APP is a very critical gene in our analyses and was the only one identified in both hub genes, mitochondrial localization genes and mitochondrial epistasis genes (Line 375-385 of page 11). The association between APP and ABAD has been confirmed by many studies. One study used a yeast two-hybrid system in which ABAD interacts with Aβ and was present in the mitochondrial matrix. Immunoprecipitation studies in the brain had also shown that the formation of complexes between ABAD and Aβ in brain and mitochondrial extracts [2]. Aβ-ABAD interaction in mitochondria reduces enzymatic function, increases the production of reactive oxygen species (ROS), and affects energy consumption. This process has also been proven in AD mouse models [3,4]. Interception of the Aβ-ABAD interaction might be a promising AD therapeutic strategy [2,5]. Taken together, our study also confirmed the indirect association of ABAD with mitochondria/APP. Based on your inspiring suggestions, we added these contents to the APP-related discussion section (Line 499-508 of page 17). We will pay more attention to this mechanism in our subsequent studies.

In addition, we appreciate your expertise in pointing out the issue of gender differences in the pathogenesis of AD. We strongly agree that this is a very interesting and important research hotspot. In our analysis, there were some mitochondria-related genes enriched in WGCNA modules, such as darkred, royalblue, darkturquoise, and plum2, that showed the most significant association with gender (Table S9). Unfortunately, due to the limitations of AD public data, we were unable to obtain phenotypic data related to female estrogen levels. Therefore, we could not explore the mechanism between reduced hormone levels and AD susceptibility based on the available data. We shall search for suitable data to investigate this topic in our further efforts. However, based on the available data, we have done some exploratory investigation to address the issue. The transcriptional mechanisms of top 10 hub genes (Table S13) in human brain development between different genders were explored using BrainSpan (http://www.brainspan.org/) and GenTree (http://gentree.ioz.ac.cn/index.php) [6] databases. It can be seen that the expression levels of these genes differ significantly between the genders at different developmental stages. For example, the expression levels of TCL1AM, RANBP10 and EPB41 peaked in the female subcortical region at 34 post-conception week (PCW) and were significantly higher than that of males at the same period. With aging, the expression levels of RASSF2, RABP10, REPS2 and TMCC3 were higher in males than in females in the same brain tissue regions, and the differences tended to increase. These results may help to explore the gender differences in the onset of AD. We have included the relevant descriptions in our paper (Line 548-557 of page 17-18, Figure S5).

In summary, we appreciate this reviewer’s constructive suggestions that are highly valuable to our future research.

[1] Rouillard AD, Gundersen GW, Fernandez NF, et al. The harmonizome: a collection of processed datasets gathered to serve and mine knowledge about genes and proteins. Database (Oxford). 2016;2016:baw100. Published 2016 Jul 3. doi:10.1093/database/baw100

[2] Chen X, Stern D, Yan SD. Mitochondrial dysfunction and Alzheimer's disease. Curr Alzheimer Res. 2006;3(5):515-520. doi:10.2174/156720506779025215

[3] Yao J, Du H, Yan S, et al. Inhibition of amyloid-beta (Abeta) peptide-binding alcohol dehydrogenase-Abeta interaction reduces Abeta accumulation and improves mitochondrial function in a mouse model of Alzheimer's disease. J Neurosci. 2011;31(6):2313-2320.

[4] Xiao X, Chen Q, Zhu X, Wang Y. ABAD/17β-HSD10 reduction contributes to the protective mechanism of huperzine a on the cerebral mitochondrial function in APP/PS1 mice. Neurobiol Aging. 2019;81:77-87.  doi:10.1016/j.neurobiolaging.2019.05.016

[5] Lim YA, Grimm A, Giese M, et al. Inhibition of the mitochondrial enzyme ABAD restores the amyloid-β-mediated deregulation of estradiol. PLoS One. 2011;6(12):e28887. doi:10.1371/journal.pone.0028887

[6] Shao Y, Chen C, Shen H, et al. GenTree, an integrated resource for analyzing the evolution and function of primate-specific coding genes. Genome Res. 2019;29(4):682-696. doi:10.1101/gr.238733.118

Reviewer 2 Report

This is an excellent paper and I have no queries regarding the genetic thrust of the paper.

Nevertheless, certain aspects of MCI-Alzheimers-control genetic material-symptom profiles would contribute to a wider readership.

Author Response

We sincerely thank this reviewer for reading our paper carefully and giving the above positive comments. Your comments are all valuable and very helpful for revising and improving our paper, as well as the important guiding significance to our researches.

We appreciate and agree with the reviewer’s insightful comment that it would be more meaningful to include research on MCI-AD-control genetic material-symptom profiles. We are deeply aware of MCI is a vital stage of cognitive remodeling and neuroplasticity. Identifying patients with MCI who are more likely to progress to AD is a key step in AD prevention. However, since most of the independent validation datasets we used, such as GSE5281, AlzData, and scREAD, contained only two phenotypes, AD and cognitive normal (CN). Therefore, only genetic variability and model predictions regarding the differences between AD and CN were highlighted in our study. In addition, for AD and MCI controls, we performed differentially expressed genes (Table S6) and mitochondria-related epistasis analysis (Line 147-160 of page 5, Line 284-371 of page 8-9) based on ADNI samples. To further complement our study for MCI-AD, we searched for literature reports of 91 hub genes related to MCI based on the PubMed. 15 hub genes have MCI-related research records, including four genes with more than ten publications (APP, GCA, PICALM, and CD22) (Line 147-160 of page 5, Line 284-371 of page 8-9). The distinction between the two gene sets is that CD22, which is known to control immune system activation, has also been identified as a negative regulator of microglia phagocytosis in the brain. CD22 inhibition improves cognitive impairment in aged mice by increasing the microglial phagocytosis of Aβ oligomers [1]. According to a recent study, higher plasma soluble CD22 (sCD22) levels are linked to faster cognitive decline. Furthermore, during 7.5 years of follow-up, plasma sCD22 levels was proven to be able to predict longitudinal cognitive decline [2]. These researches indicated that CD22 may be a potential biomarker for identifying people who are more prone to progress from MCI to AD. It also suggests that our study may be able to provide some resolution of the pathological mechanisms underlying the transformation of MCI to AD. Based on these findings, we updated the relevant parts of the manuscript (Line 511-520 of page 17). Hopefully, our study can provide some information for your rigorous consideration to suit a wider range of researchers. In the meantime, we are working on some specific aspects about the mitochondria-related genetic transition between AD-MCI-CN. Your comments are very valuable and have inspired us greatly.

Besides, the revised manuscript has been thoroughly checked and the grammatical errors and typos have been corrected. Thank you again for your insightful suggestions to make our paper more informative.

[1] Pluvinage JV, Haney MS, Smith BAH, et al. CD22 blockade restores homeostatic microglial phagocytosis in ageing brains. Nature. 2019;568(7751):187-192. doi:10.1038/s41586-019-1088-4

[2] Bu XL, Sun PY, Fan DY, et al. Associations of plasma soluble CD22 levels with brain amyloid burden and cognitive decline in Alzheimer's disease. Sci Adv. 2022;8(13):eabm5667. doi:10.1126/sciadv.abm5667